# Prognostic Role of Serum Procalcitonin Measurement in Adult Patients Admitted to the Emergency Department with Fever

**DOI:** 10.3390/antibiotics10070788

**Published:** 2021-06-29

**Authors:** Marcello Covino, Alberto Manno, Giuseppe De Matteis, Eleonora Taddei, Luigi Carbone, Andrea Piccioni, Benedetta Simeoni, Massimo Fantoni, Francesco Franceschi, Rita Murri

**Affiliations:** 1Emergency Medicine, Fondazione Policlinico Universitario A. Gemelli, IRCSS, 00168 Rome, Italy; marcello.covino@unicatt.it (M.C.); alberto.manno@policlinicogemelli.it (A.M.); luigi.carbone@policlinicogemelli.it (L.C.); andrea.piccioni@policlinicogemelli.it (A.P.); benedetta.simeoni@policlinicogemelli.it (B.S.); francesco.franceschi@unicatt.it (F.F.); 2Fondazione Policlinico Universitario Agostino Gemelli IRCCS, Università Cattolica del Sacro Cuore, 00168 Rome, Italy; massimo.fantoni@policlinicogemelli.it (M.F.); Rita.Murri@unicatt.it (R.M.); 3Department of Internal Medicine, Fondazione Policlinico Universitario A. Gemelli, IRCSS, 00168 Rome, Italy; 4Department of Infectious Diseases, Fondazione Policlinico Universitario A. Gemelli IRCCS, 00168 Rome, Italy; eleonora.taddei@policlinicogemelli.it

**Keywords:** procalcitonin, qSOFA, sepsis, fever, antibiotic treatment

## Abstract

**Background and Objectives.** Fever is one of the most common presenting complaints in the Emergency Department (ED). This study aimed at evaluating the prognostic role of serum Procalcitonin (PCT) measurement among adult patients admitted to the ED with fever. **Materials and Methods.** This is a retrospective cross-sectional study including all consecutive patients admitted to ED with fever and subsequently hospitalized in a period of six-year (January 2014 to December 2019). Inclusion criteria were age > 18 years, fever (T ≥ 38 °C) or chills within 24 h from presentation to the ED as the main symptom, and availability of a PCT determination obtained <24 h since ED access. The primary endpoint was overall in-hospital mortality. **Results.** Overall, 6595 patients were included in the study cohort (3734 males, 55.6%), with a median age of 71 [58–81] years. Among these, based on clinical findings and quick sequential organ failure assessment (qSOFA), 422 were considered septic (36.2% deceased), and 6173 patients non-septic (16.2% deceased). After correction for baseline covariates, a PCT > 0.5 ng/mL was an independent risk factor for all-cause in-hospital death in both groups (HR 1.77 [1.27–2.48], and 1.80 [1.59–2.59], respectively). **Conclusions.** Among adult patients admitted with fever, the PCT assessment in ED could have reduced prognostic power for patients with a high suspicion of sepsis. On the other hand, it could be useful for sepsis rule-out for patients at low risk. In these latter patients, the prognostic role of PCT is higher for those with a final diagnosis of bloodstream infection.

## 1. Introduction

Fever is one of the most frequent symptoms reported by patients evaluated in the Emergency Department (ED) [1]. Indeed, it may suggest the presence of a broad spectrum of diseases, ranging from self-limiting inflammatory disorders to potentially life-threatening conditions such as sepsis [2]. 

According to the Third International Consensus Definitions for Sepsis and Septic Shock (Sepsis-3), sepsis is defined as a life-threatening organ dysfunction caused by a dysregulated host response to infection [3]. In settings outside the Intensive Care Unit (ICU), as the ED, quick sequential organ failure assessment score (qSOFA) is considered a simple and reliable tool to identify patients who are likely to be septic [3]. The qSOFA was preferred over a full SOFA score assessment since it is widely used in the ED setting for the evaluation and the risk stratification of patients with fever [3]. A qSOFA score ≥ 2 should prompt the emergency physician to investigate for organ dysfunction, evaluate the initiation of appropriate antibiotic therapy and increase the frequency of monitoring of such patients. 

To date, several efforts were carried on to find an association between biomarkers and prognosis in septic patients [4,5,6,7,8,9]. Assicot et al. in 1993 first described the association of increased PCT, a pro-hormone of human calcitonin with the severity of infection in children [10]. Since then, several studies have described the possible role of PCT as a prognostic marker for sepsis. Nevertheless, although the International Guidelines for Management of Sepsis and Septic shock of 2016 recommend the use of PCT as a prognostic biomarker for sepsis [11], evidence from the literature is still contradictory [2,5,6,12,13,14,15,16,17,18,19,20]. Moreover, in the last few years, several studies have suggested that PCT may play a role in the prognostic assessment of all critically ill patients regardless of the presence of septic complications [16,21,22].

The present study aimed at assessing the prognostic role of the serum PCT measured at the ED admission in adult patients evaluated for fever and admitted to a ward.

## 2. Materials and Methods

### 2.1. Study Design

This was a cross-sectional observational retrospective study conducted in the ED of a teaching hospital with an annual attendance of about 75,000 patients, 87% adults. We identified all consecutive patients admitted to the ED with fever from January 2014 to December 2019 and then hospitalized.

Inclusion criteria were age > 18 years, fever (T ≥ 38 °C) or chills within 24 h from presentation to the ED as the main symptom, and availability of a PCT determination obtained <24 h since ED access.

Exclusion criteria were age < 18 years and pregnancy.

### 2.2. Study Variables

For all patients included in the study cohort, the qSOFA score was calculated. According to the qSOFA score and clinical presentation, patients were divided into suspected sepsis (Sep group) and non-suspected sepsis (NSep group).

Vital parameters were obtained at ED admission. In the case of several measurements, the first values were considered.

Patients’ comorbidities were recorded including severe obesity (body mass index ≥ 40), hypertension, ischemic heart disease, heart failure, chronic respiratory obstructive disease (COPD), peripheral vascular disease, dementia, diabetes, chronic kidney disease, malignancy, and leukemia/lymphoma. Overall comorbidity status was assessed by the Charlson comorbidity index [23].

All patients had a blood sampling for routine laboratory testing and PCT value determination. Blood cultures were obtained based on emergency physician judgment. The cut-off value of PCT serum level predictive of sepsis was set at 0.5 ng/mL. All Testing was available 24 h a day in our ED.

### 2.3. Outcome Measures

The primary endpoint of the study was the all-cause in-hospital mortality.

### 2.4. Statistical Analysis

Categorical variables are presented as absolute numbers and percentages; continuous variables are presented as median (interquartile range). Categorical variables were statistically compared by Chi-square test or Fisher exact test as appropriate. Continuous variables were compared by the Mann–Whitney U test.

Significant factors at univariate analysis were entered into a Cox regression model to identify independent risk predictors for the defined outcomes. PCT value > 0.5 ng/mL was forced in all the models. Survival curves were calculated according to the Kaplan–Meier method. Cox regression analysis results were reported as Hazard Ratio (HR) [95% confidence interval].

Univariate and multivariate analyses of factors possibly influencing prognosis in terms of in-hospital all-cause mortality, including the PCT value at presentation in the ED, were made separately in Sep and NSep groups.

In the NSep group, a further sub-analysis assessed the possible correlation of ED PCT and prognosis in patients with different infective disease diagnoses at discharge.

Receiver operating characteristics (ROC) curve analysis was used to assess the relationship between different PCT values and all-causes in-hospital death, and to determine the sensitivity and specificity for the defined outcome at different PCT cut-off levels.

*p* values were 2-sided, with a significance threshold of 0.05, and corrected in case of multiple group comparison. The study analysis was conducted by SPSS version 25 (IBM, Armonk, NY, USA).

## 3. Results

In the study period, 33,468 adults were evaluated for fever in our ED, and 8183 (24.4%) were subsequently admitted in-hospital. Among these, 6595 patients had a PCT sampling in ED and were included in the study cohort.

There were 2861 females (44.4%) and 3734 males (55.6%), with a median age of 71 (58–81) years. According to the qSOFA score and clinical presentation, 422 patients (6.4%) were suspected of sepsis [Sep group] while 6173 (96.6%) did not meet the criteria for suspected sepsis (NSep group).

### 3.1. Patients Characteristics

Patients with suspected sepsis at presentation were significantly older, mostly females, and had higher PCT values at admission (Table 1). The most frequent complaints recorded included fever, dyspnea, abdominal pain, and vomiting. Vital parameters reflected the derangement in vital signs as also demonstrated by qSOFA.

Interestingly, a septic presentation was more frequent in patients with a higher number of comorbidities as expressed by CCI. Patients with septic presentation showed more frequently a medical history of heart failure, peripheral vascular disease, dementia, diabetes, COPD, kidney disease, and malignancy.

As expected, septic presentation patients showed a worse outcome both in terms of the need for mechanical ventilation, and all-cause in-hospital death. An infective diagnosis at discharge was made for about 2/3 of patients in the septic group, as compared to about ½ in the non-septic presentation.

Detailed demographic and clinical data of the study patients are shown in Table 1.

### 3.2. In-Hospital Mortality in Patients with Non-Suspected Sepsis (NSep Group)

Among the 6173 patients with a non-septic presentation at the ED admission, 1001 (16.2%) died.

At univariate analysis, a PCT > 0.5 ng/mL was significantly associated with in-hospital death. Moreover, we found that deceased patients were older, and reported more fever, dyspnea, abdominal pain, vomiting, and asthenia/malaise at the ED visit (Table 2). As expected, most of the physiological parameters were significantly different in the deceased group, except for heart rate (Table 2). Non-survivors also had more comorbidities, particularly cardiovascular diseases (Table 2).

Multivariate Cox regression analysis demonstrated that once corrected for clinical, demographical and comorbidity covariates, a PCT > 0.5 was an independent risk factor for all-cause in-hospital death in these patients, with about a two-fold risk when compared to PCT ≤ 0.5 ng/mL (HR 1.80 [1.59–2.59], *p* < 0.001) (Table 2).

### 3.3. Patients with Suspected Sepsis at ED Admission (Sep Group)

In patients with high suspicion of sepsis at presentation, a higher respiratory rate and a lower diastolic blood pressure were found to be associated with death. (Table 2). As expected, also in this group a higher number of comorbidities, as expressed by higher CCI were associated with a worse prognosis (Table 3).

Multivariate Cox regression analysis demonstrated that once corrected for significant clinical, demographical and comorbidity covariates, a PCT > 0.5 ng/mL was an independent risk factor for all-cause in-hospital death also in the Sep group (HR 1.77 [1.27–2.48], *p* = 0.001). Interestingly, the hazard ratio for PCT > 0.5 ng/mL was similar for Sep and NSep groups (Table 2 and Table 3).

### 3.4. Sub-Analysis of PCT Prognostic Value According to Discharge Diagnosis

In patients with a non-septic presentation in the ED (NSep group), a PCT > 0.5 ng/mL was independently associated with overall in-hospital death, and this was particularly evident in patients with infective diagnosis at discharge (Table 4).

The association between PCT > 0.5 ng/mL and death for infective diagnosis was different according to the site of infection. Patients with elevated PCT at admission had a significantly higher death rate if affected by bloodstream infections, and abdominal infections. Conversely, this association was absent or countertrend for urinary tract infections, and pneumonia, respectively. However, the reduced sample in this fragmented evaluation could indeed affect the statistical power of our analysis.

As a whole, a PCT > 0.5 ng/mL was associated with death in most diagnoses, but the association was particularly strong for bloodstream infections (Table 4).

### 3.5. ROC Curve Analysis of PCT Value for All-Cause In-Hospital Death

Overall, PCT at admission had a good discrimination power for all-cause in-hospital death. Interestingly the discrimination was higher in non-septic patients than in patients with suspected sepsis at ED presentation (Table 5 and Table 6). Overall ROC AUC was 0.607 [0.595–0.619] in the NSep group, and was 0.580 [0.532–0.628] in the Sep group. As showed by the ROC graph in Figure 1, the PCT value in ED had a direct ascending correlation with poor outcome in the NSep group, with high negative predictive values (NPV) for low PCT values (Figure 1, Table 5).

Conversely, in the Sep group, low PCT values had a poor discrimination ability, and consequently a low NPV. Only for elevated PCT values, the PCT showed a direct correlation with all-cause in-hospital death, with an 80% specificity achieved only for PCT > 7.5 ng/Ml (Figure 1).

## 4. Discussion

The main finding of the present study is that among adult patients admitted to the ED with fever and subsequently hospitalized, an elevated PCT value at admission is an independent risk predictor for death both in septic patients and in patients with a non-septic presentation.

To date, the association of a high level of PCT with a poor prognosis has been widely analyzed in several clinical contexts [5,8,13,14,15,16,17,18,19,20]. At the same time, the possible correlation between a high level of PCT and prognosis may have a crucial role in the management of patients with fever of possible infective origin, particularly in ED [24,25,26,27,28,29,30]. Indeed, the early identification of patients at risk for sepsis allows both precocious directed therapy and specific intensive care management, significantly improving clinical outcomes [31].

Moreover, the association between changes in PCT level during the disease and the progression of sepsis was also described. A reduction of more than 30% of the PCT value in the first 24 h following the onset of antibacterial treatment may reflect the appropriateness of therapy and the control of sepsis [2]. Consequently, PCT levels variability is used by antibiotic stewardship teams to optimize therapy during hospitalization [7,32,33,34,35,36].

While PCT has a confirmed clinical utility both in risk stratification and in antibiotic management [12,16,17,18,19,20,32,33,34,35,36,37], its relevance for patient’s level outcomes is still debated. Considering ICU patients, recent meta-analyses seem to confirm the positive impact of PCT-guided management on survival [15,36]. In particular, Wirz Y et al. in their meta-analysis including ICU septic patients showed that PCT-guided antibiotic treatment resulted in improved survival and shorter antibiotic treatment duration [36].

The prognostic relevance of the PCT determination at ED admission is still uncertain [6,24,25,26,27,28,29,38]. Among the 25 studies included in the meta-analysis by Arora et al., only four were conducted in ED, and in three of them, the PCT level on day 1 was significantly different between surviving and non-surviving septic patients [17]. Similarly, Hua Y et al. confirmed the prognostic role of PCT assessment in the ED even in a septic population [38].

In our series, only 422 out of the 6595 feverish patients could be defined as septic at ED admission according to Sepsis-3 criteria. Not surprisingly, in these patients, a PCT value > 0.5 ng/mL was an independent risk factor for in-hospital mortality (HR 1.77 [1.27–2.48]; *p* < 0.001).

However, the vast majority of patients currently admitted to EDs with fever do not present a septic status. Furthermore, many patients accessing the ED are already taking antipyretics or other prescribed therapy. As a consequence, the utility of serum PCT determination in all patients admitted to ED with fever is still a matter of debate [7,8,13,14,19,39,40,41]. In our cohort, >95% of patients were not classified as having suspected sepsis. Nevertheless, also in this group a PCT > 0.5 ng/mL at ED admission was associated with all-cause in-hospital death, even after adjusting for medical history, demographic, and clinical presentation at the ED (HR 1.80 [1.59–2.59]; *p* < 0.001). The correlation of serum PCT value with mortality could in part also be explained by the significant association (Table 2), already demonstrated in the literature, between PCT and medical history of heart failure [42].

Although a high PCT was associated with worse survival in the NSep group, only 56% of these patients had a discharge diagnosis of infectious disease. In this latter group, a PCT > 0.5 ng/mL was associated with death in most infective conditions, and the association was particularly evident in patients with bloodstream infection (Table 4). Nevertheless, the usefulness of early PCT assessment in the ED for patients at low risk for sepsis remains unclear [39]. Several reports investigating the role of early PCT assessment in the ED for feverish but not septic patients affected by pneumonia [8], urinary tract infection [7], or intra-abdominal infection [14], confirmed some prognostic usefulness only for patients aged ≥ 85 years, or with severe comorbidities, or with a diagnosis of bloodstream infection. On the other hand, the studies conducted in the ED setting on lower respiratory tract infections [35,40] and urinary tract infections [41] did not highlight an association between early PCT determination and a better patient outcome.

In patients with non-suspected sepsis at ED presentation, the PCT value showed a direct correlation with a higher risk of in-hospital death. However, the overall discrimination ability of PCT was reduced, with a ROC AUC 0.607 [0.595–0.619]. Interestingly, in these patients, a low PCT had a fair/good NPV, and patients with PCT < 0.05 had a >90% chance of survival. This is of uttermost relevance in the ED setting, where the ED physician could be confident to discharge patients with low PCT who are at low evolutive risk. Nevertheless, as confirmed by the available literature, the accuracy of PCT alone is too low to use it as an exclusive predictor of poor outcome [43,44].

In the group of patients with suspected sepsis at ED presentation, the PCT value showed a reduced ability in discriminating patients at risk for in-hospital death, and a reduced NPV. Furthermore, the discriminating ability was quite reduced for low PCT values, as showed by the ROC curve (Figure 1). These results confirm the meta-analyses showing that PCT may not be useful as a single indicator for prognostic stratification in the overt septic patient [42], and could be not superior to the available clinical prognostic indexes in use for patients with community-acquired pneumonia [43]. Nonetheless, the specific diagnostic accuracy of PCT in the ED setting remains unknown, because of the limited number of available studies [42].

As a result, the PCT value in ED should be considered only as an addition to relevant clinical parameters, and the antibiotic prescription should be supervised by an antibiotic stewardship team. In this way, the PCT could be pivotal for quickly starting the antibiotic therapy only in patients who could truly benefit from these drugs, avoiding prolonged or useless treatment courses.

### Study Limitations

As for any retrospective study, some limitations are worth considering. The PCT determination was requested at the discretion of the treating physician providing a selection bias that cannot be avoided in our analysis. At the same time, patients were included in the analysis regardless of whether they had already started therapy before accessing the ED. Furthermore, this is a monocentric study conducted in a large teaching hospital, and our results could not be generalizable to all EDs. This latter limitation is counterbalanced by the uniformity and consistency of the hospital treatments that were provided by the same antibiotic stewardship team for all the patients in the study cohort. Finally, we did not measure the PCT sampling during the in-hospital stay, and cannot provide an evaluation about the role of the further PCT kinetics on patients’ outcomes.

## 5. Conclusions

Among adult patients admitted to ED with fever, the PCT measurement could represent a prognostic tool for the risk stratification, and an efficient biomarker for the rule-out of a sepsis diagnosis, particularly in those with low pre-test suspicion.

However, although in the vast majority of feverish adults with a non-septic presentation, apart from those with a final diagnosis of bloodstream infection, the overall magnitude of its role remains uncertain. As a result, the antibiotic stewardship teams should use the ED PCT value to guide treatment only in addition to relevant clinical parameters.

Further multicentric prospective studies in ED populations are needed to confirm and expand these results.

## Figures and Tables

**Figure 1 antibiotics-10-00788-f001:**
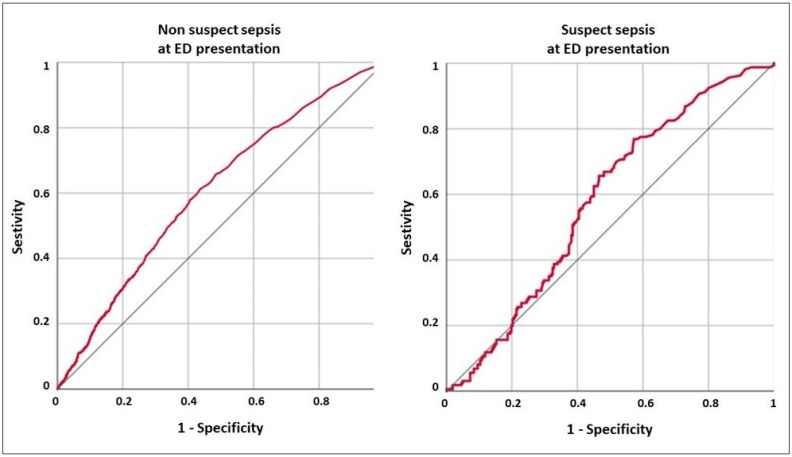
Graph of ROC curve analysis for PCT values in ED) for all-cause in-hospital death, both for non-suspected and suspected sepsis in ED (NSep and Sep groups). In the NSep group, the raising of PCT value was associated with a progressive increase in specificity. In the Sep patients, a low value of PCT had poor specificity and sensitivity for in-hospital death, and a specificity >80% was achieved only for PCT value >7.5 ng/mL.

**Table 1 antibiotics-10-00788-t001:** Demographic and clinical characteristics of the patients included in the study. Values are shown separately for patients with suspected sepsis (Sep group) and non-suspected sepsis (NSep) at Emergency Department admission.

Variable	All Patients *n* = 6595	NSep *n* = 6173	Sep *n* = 422	*p*
Age	71 [58–81]	71 [58–81]	79 [68–86]	<0.001
Sex (Male)	3734 (55.6)	3518 (57.0)	216 (51.2)	0.020
PCT > 0.5 ng/mL	2866 (43.5)	2630 (42.6)	236 (55.9)	<0.001
PCT value	0.38 [0.14–1.78]	0.36 [0.14–1.65]	0.71 [0.23–5.61]	<0.001
Emergency Department Presentation				
Dyspnea	1571 (23.8)	1370 (22.2)	201 (47.6)	<0.001
Cough	442 (6.7)	416 (6.7)	26 (6.2)	0.646
Abdominal pain	1051 (16.7)	1028 (16.6)	23 (5.5)	<0.001
Vomiting	704 (10.7)	676 (11.0)	28 (6.6)	0.005
Diarrhea	425 (6.4)	403 (6.5)	22 (5.2)	0.287
Neurological symptoms	285 (4.3)	272 (4.4)	13 (3.1)	0.195
Syncope	296 (4.5)	276 (4.5)	20 (4.7)	0.797
Malaise/asthenia	851 (12.9)	806 (13.1)	45 (10.7)	0.156
Vital Parameters				
Heart rate	94 [80–110]	94 [80–110]	93 [76–111]	0.279
Respiratory rate	19 [15–20]	19 [15–20]	21 [20–25]	<0.001
GCS	15 [15–15]	15 [15–15]	14 [13–15]	<0.001
Systolic BP	122 [107–140]	123 [110–140]	90 [80–115]	<0.001
Diastolic BP	72 [61–82]	73 [63–83]	60 [50–70]	<0.001
SaO_2_	96 [93–98]	96 [93–98]	89 [85–92]	<0.001
Comorbidities				
Charlson Index	5 [3–7]	5 [3–7]	6 [4–8]	<0.001
Severe obesity	135 (2.0)	121 (2.0)	14 (3.3)	0.057
Hypertension	1895 (28.7)	1761 (28.5)	134 (31.8)	0.157
CAD	1102 (16.7)	1022 (16.6)	80 (19.0)	0.201
Heart failure	1250 (19.0)	1131 (18.3)	119 (28.2)	<0.001
PVD	1452 (22.0)	1326 (21.5)	126 (29.9)	<0.001
Dementia	422 (6.4)	237 (5.3)	185 (8.8)	<0.001
COPD	938 (14.2)	838 (13.6)	100 (23.7)	<0.001
Diabetes	1461 (22.2)	1344 (21.8)	117 (27.7)	0.004
CKD	1480 (22.4)	1351 (21.9)	129 (30.6)	<0.001
Leukemia/lymphoma	545 (8.3)	517 (8.4)	28 (6.6)	0.209
Malignancy	1789 (27.1)	1708 (27.7)	81 (19.2)	<0.001
Outcomes				
Mechanical Ventilation	584 (8.9)	518 (8.4)	66 (15.6)	<0.001
Death	1161 (17.6)	1001 (16.2)	160 (37.9)	<0.001
Length of hospital stay	10.6 [6.5–18.5]	10.6 [6.5–18.5]	10.9 [5.9–18.4]	0.461

Abbreviations: PCT—Procalcitonin; GCS—Glasgow Coma Scale; BP—Blood Pressure; qSOFA—quick sequential organ failure assessment; CAD—Coronary artery disease; PVD—Peripheral Vascular Disease; COPD—Chronic obstructive pulmonary disease; CKD—Chronic kidney disease.

**Table 2 antibiotics-10-00788-t002:** Univariate and multivariate analysis for all-cause in-hospital death in patients with a non-septic presentation in the Emergency Department (NSep group).

Variable	Survived *n* = 5172	Deceased *n* = 1001	Univar. *p*	Hazard Ratio	Multiv. *p*
Age	70 [56–80]	78 [68–85]	<0.001	1.03 [1.02–1.03]	<0.001
Sex (Male)	2950 (57.0)	568 (56.7)	0.863		
PCT > 0.5 ng/mL	2063 (39.9)	567 (56.6)	<0.001	1.80 [1.59–2.59]	<0.001
PCT value	0.33 [0.13–1.45]	0.73 [0.23–3.66]	<0.001		
Charlson Index	5 [3–7]	6 [4.5–8]	<0.001		
Severe obesity	102 (2.0)	19 (1.9)	0.877		
Hypertension	1498 (29.0)	263 (26.3)	0.084		
CAD	804 (15.5)	218 (21.8)	<0.001	1.11 [0.92–1.33]	0.284
Heart failure	833 (16.1)	298 (29.8)	<0.001	1.43 [1.23–1.24]	<0.001
PVD	1054 (20.4)	272 (27.2)	<0.001	0.95 [0.80–1.13]	0.587
Dementia	232 (4.5)	78 (7.8)	<0.001	1.25 [1.02–1.52]	0.028
COPD	694 (13.4)	144 (14.4)	0.413		
Diabetes	1107 (21.4)	237 (23.7)	0.111		
CKD	1045 (20.2)	306 (30.6)	<0.001	1.10 [0.96–1.27]	0.170
Leukemia/lymphoma	412 (8.0)	105 (10.5)	0.008	1.24 [1.09–1.43]	0.002
Malignancy	1410 (27.3)	298 (29.8)	0.104		

Abbreviations: PCT—Procalcitonin; CAD—Coronary artery disease; PVD—Peripheral Vascular Disease; COPD—Chronic obstructive pulmonary disease; CKD—Chronic kidney disease.

**Table 3 antibiotics-10-00788-t003:** Univariate and multivariate analysis for in-hospital death in patients with a septic presentation in the Emergency Department (Sep group).

Variable	Survived *n* = 262	Deceased *n* = 160	Univar. *p*	Hazard Ratio	Multiv. *p*
Age	78 [68–85]	82 [70–87]	0.003	1.02 [1.01–1.04]	0.003
Sex (Male)	131 (50.0)	85 (53.1)	0.533		
PCT > 0.5 ng/mL	129 (49.2)	107 (66.9)	<0.001	1.77 [1.27–2.48]	0.001
PCT value	0.49 [0.17–4.93]	1.11 [0.34–6.56]	0.006		
Charlson Index	6 [4–8]	7 [5–9]	0.003		
Severe obesity	10 (3.8)	4 (2.5)	0.582		
Hypertension	86 (32.8)	48 (30.0)	0.545		
CAD	44 (16.8)	36 (22.5)	0.147		
Heart failure	63 (24.0)	56 (35.0)	0.015	1.34 [0.95–1.88]	0.097
PVD	74 (28.2)	52 (32.5)	0.354		
Dementia	36 (13.7)	16 (10.0)	0.257		
COPD	62 (23.7)	38 (23.8)	0.984		
Diabetes	63 (24.0)	54 (33.8)	0.031	1.25 [0.89–1.74]	0.187
CKD	69 (26.3)	60 (37.5)	0.016	1.21 [0.87–1.69]	0.264
Leukemia/lymphoma	17 (6.5)	11 (6.9)	0.877		
Malignancy	47 (17.9)	34 (21.3)	0.402		

Abbreviations: PCT—Procalcitonin; CAD—Coronary artery disease; PVD—Peripheral Vascular Disease; COPD—Chronic obstructive pulmonary disease; CKD—Chronic kidney disease.

**Table 4 antibiotics-10-00788-t004:** Outcomes of patients in the study cohort for non-septic presentation in ED (qSOFA <2) according to PCT value at admission and discharge diagnosis.

Variable	Total Deceased	Deceased in PCT ≤ 0.5 ng/mL	Deceased in PCT > 0.5 ng/mL	*p* Value
**Non-Septic** **Presentation in the ED**	***n* = 1001**	***n*** = **434**	***n* = 567**	
Pneumonia	235 (23.5)	116 (26.7)	119 (21.0)	0.034
Abdominal infection	67 (6.7)	20 (4.6)	47 (8.3)	0.021
Urinary tract infection	55 (5.5)	25 (5.8)	30 (5.3)	0.747
Bloodstream infection	346 (34.6)	110 (25.3)	236 (41.6)	<0.001
Other infections	40 (4.0)	19 (4.4)	21 (3.7)	0.589
Any infective diagnosis	574 (57.3)	225 (51.8)	349 (61.6)	0.002

**Table 5 antibiotics-10-00788-t005:** Sensitivity and specificity for different cut-off values of PCT at Emergency Department admission for non-septic presentation in ED (NSep group), for all-cause in-hospital death. Overall ROC AUC was 0.607 [0.595–0.619]. The best discriminating value according to Youden index J was PCT > 0.42 ng/mL.

Cut-Off Value	Sensitivity	Specificity	Positive Predictive Value (PPV)	Negative Predictive Value (NPV)
>0.05 ng/mL	98.2 [97.2–98.9]	4.3 [3.8–4.9]	16.6 [16.4–16.7]	92.5 [88.5–95.2]
>0.5 ng/mL	56.6 [53.5–59.7]	60.1 [58.8–61.5]	21.6 [20.5–22.7]	87.8 [86.9–88.5]
>1 ng/mL	42.6 [39.6–45.8]	71.2 [69.9–72.4]	22.3 [20.9–23.8]	86.5 [85.8–87.2]

**Table 6 antibiotics-10-00788-t006:** Sensitivity and specificity for different cut-off values of PCT at emergency department admission for suspected sepsis in ED (Sep group), for all-cause in-hospital death. Overall ROC AUC was 0.580 [0.532–0.628). The best discriminating value according to Youden index J was PCT > 0.33 ng/mL.

Cut-Off Value	Sensitivity	Specificity	Positive Predictive Value (PPV)	Negative Predictive Value (NPV)
>0.05 ng/mL	98.7 [95.6–100.0]	1.1 [0.2–3.3]	37.9 [37.4–38.4]	60.0 [20.2–89.9]
>0.5 ng/mL	66.9 [59.0–74.1]	50.8 [44.5–57.0]	45.3 [41.3–49.4]	71.5 [66.1–76.3]
>1 ng/mL	51.2 [43.2–59.2]	61.1 [54.9–67.0]	44.6 [39.4–49.9]	67.2 [63.0–71.2]

## Data Availability

The data presented in this study are available upon a reasonable request from the corresponding author.

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
