# Peer review of "Prognostic Role of Serum Procalcitonin Measurement in Adult Patients Admitted to the Emergency Department with Fever"

_antibiotics, 2021, doi:10.3390/antibiotics10070788_

Round 1

Reviewer 1 Report

General comments

The abstract should be written in a single paragraph without the headings

While the study is well reported, the discussion section is shallow. The authors need to give more details on the results they obtained. For example, what was the impact of age or comorbidities on their observations. Line 197-219 is more of literature review than discussion.

Patients who came in to the ED already on some form of treatment should have been excluded from the study as this could affect the overall outcome. How do the authors account for this?

Specific comments

Line 16: Delete “was”. This study aimed at evaluating…..

Line 18: Delete “and then hospitalized”. Similarly, this appears in Line 20-21. I stand to be corrected; to hospitalise means “admit or cause (someone) to be admitted to hospital for treatment”. So, I wonder why the authors say “admitted and subsequently hospitalized”

Line 62: The present study aimed at assessing…

Line 67: This was…

Line 81: Delete “were available”

Line 89: testing

Line 120: Delete “considered”

Line 124-125: ….older, mostly females,….. The most frequent complaints recorded included fever, ……

Line 183-184: Nevertheless,….in others” please rephrase this section.

Reviewer 2 Report

The topic of this manuscript is very interesting. The reviewer feels the manuscript can be accepted after some minor amendments.

(1) Any sub-group analysis for gender, age, type of infection? (The sample size may be too small to do so).

(2) Any more detailed statistical analysis correlation of exact values of PCT and outcome?
